# The Pharmacological Potential of Adenosine A_2A_ Receptor Antagonists for Treating Parkinson’s Disease

**DOI:** 10.3390/molecules27072366

**Published:** 2022-04-06

**Authors:** Akihisa Mori, Jiang-Fan Chen, Shinichi Uchida, Cecile Durlach, Shelby M. King, Peter Jenner

**Affiliations:** 1Kyowa Kirin Co., Ltd., Tokyo 100-0004, Japan; akihisamori.23@gmail.com (A.M.); shinichi.uchida.fd@kyowakirin.com (S.U.); 2Molecular Neuropharmacology Laboratory, Wenzhou Medical University, Wenzhou 325015, China; chenjf555@gmail.com; 3Kyowa Kirin International, Galashiels TD1 1QH, UK; cecile.durlach@kyowakirin.com; 4Kyowa Kirin Inc., Bedminster, NJ 07921, USA; shelby.king.b3@kyowakirin.com; 5Institute of Pharmaceutical Science, Kings College London, London SE1 9NH, UK

**Keywords:** adenosine, A_2A_ receptors, G protein coupled receptor, istradefylline, non-dopaminergic target, Parkinson’s disease

## Abstract

The adenosine A_2A_ receptor subtype is recognized as a non-dopaminergic pharmacological target for the treatment of neurodegenerative disorders, notably Parkinson’s disease (PD). The selective A_2A_ receptor antagonist istradefylline is approved in the US and Japan as an adjunctive treatment to levodopa/decarboxylase inhibitors in adults with PD experiencing OFF episodes or a wearing-off phenomenon; however, the full potential of this drug class remains to be explored. In this article, we review the pharmacology of adenosine A_2A_ receptor antagonists from the perspective of the treatment of both motor and non-motor symptoms of PD and their potential for disease modification.

## 1. Introduction

The identification, characterization and classification of adenosine receptors have opened new vistas in pharmacology for treating a wide range of peripheral and central disease states [1,2,3,4,5]. This is particularly true of the potential use of centrally acting adenosine A_2A_ receptor antagonists for the treatment of neurological and psychiatric illnesses where current therapies provide incomplete symptom relief and where there are no proven strategies that slow or prevent disease progression [6,7,8,9].

Adenosine receptors are widely expressed throughout the central nervous system and peripheral tissues or cells (for recent reviews, see Chen, 2014 [7]; Mori 2021 [10]) [7,10,11,12,13,14,15,16,17,18,19,20]. They belong to the family of G protein-coupled receptors (GPCRs), which possess seven transmembrane alpha helices, with three intracellular loops and three extracellular loops [11,21]. Adenosine receptors were initially cloned in the 1970s and 1980s, and then classified into four receptor subtypes based on their pharmacological profiles to inhibit (A_1_, A_3_) or stimulate (A_2A_, A_2B_) adenylate cyclase (AC) activity [11]. In particular, adenosine A_2A_ receptors have a highly selective localization in brain [12] and are one of the best-characterized GPCRs [22,23]. As discussed in detail below, the selective localization of A_2A_ receptors to the basal ganglia and the limbic brain region together with their ability to modulate the function of a range of other neurotransmitter systems offers a non-dopaminergic pharmacological approach to the treatment of Parkinson’s disease (PD), and potentially common neuropsychiatric disorders, such as depression and cognitive decline [4,6,9]. In addition, the positioning of A_2A_ receptors on both neurons and glial cells and the ability to modulate pathogenic processes such as excessive glutamate release and the aggregation of toxic protein species identifies them as relevant to providing neuroprotective or disease modifying therapies in a range of neurodegenerative illnesses [17,20]. This has been demonstrated by the ability of A_2A_ receptor antagonists to prevent neuronal death in preclinical models relevant to PD, Alzheimer’s disease (AD) and Huntington’s chorea [24,25,26,27].

In the following sections, we will describe in detail the rationale for the use of adenosine A_2A_ antagonists, notably istradefylline for the treatment of PD, as well as the potential for treatment of a range of neuropsychiatric disorders and for application as a neuroprotective strategy. In addition, we will briefly expand on the range of degenerative disease states that might also be targets for therapy with A_2A_ receptor antagonists.

## 2. Adenosine A_2A_ Receptors and Parkinson’s Disease

To date, the exploitation of the central pharmacology of the A_2A_ receptor has centered on the use of antagonist molecules for the treatment of PD, and for that reason, it will form the backbone of this review. However, since PD is characterized by both motor and non-motor symptoms, including a range of neuropsychiatric symptoms (depression, cognitive decline, sleep disturbance), this may hint at the potential of adenosinergic modulation as a target for central nervous system disorders in general [28,29,30]. A limitation with the current symptomatic therapies for PD is that they are almost entirely based on dopamine-replacement therapy (levodopa, dopamine agonists, enzymatic inhibitors) that shows a loss of efficacy (response fluctuations such as ‘wearing off’) with disease duration and severity, and where significant side effects occur (dyskinesia, hallucinations, impulse control disorders, sleep disturbance) [31]. In addition, most non-motor symptoms are only partially responsive to dopaminergic drugs, leaving a high degree of unmet need [8]. However, the widespread adaptive changes that occur within the basal ganglia as a result of the loss of dopaminergic neurons affect multiple neuronal systems that are non-dopaminergic in nature, offering potential new targets for treating both motor and non-motor symptoms [30,32].

In terms of other non-dopaminergic approaches, only the non-specific and weak N-methyl-D-aspartate (NMDA) antagonist amantadine and anticholinergic drugs such as benzhexol have been used in humans to treat motor symptoms, and both have undesirable adverse effects in PD [33,34]. More recently, the 5-HT_2A_ inverse agonist pimavanserin has been approved for the treatment of psychosis in PD, supporting the potential for non-dopaminergic approaches [35]. However, despite considerable preclinical and clinical investigation, most non-dopaminergic drugs examined in clinical trials for improving motor symptoms in PD (serotoninergic drugs, noradrenergic antagonists, glutamate antagonists) have failed to show efficacy or have shown undesirable side effects [36,37]. It is for this reason that adenosine A_2A_ receptor antagonists are of such interest as they may offer an alternative for treatment of motor and non-motor symptoms balanced with an acceptable adverse event profile and the potential for disease modifying activity.

### 2.1. Basal Ganglia Expression of A_2A_ Receptors

The limited distribution of A_2A_ receptors to the striatum, external globus pallidus (GPe), nucleus accumbens, and olfactory tubercle [38,39,40,41] has important implications for the treatment of PD as the most commonly occurring basal ganglia disorder [12,20,42]. Normal motor function is mediated in the striatum by the balance in activity between the direct and indirect GABAergic striatal output pathways, which in turn alters the activity of the striato-thalamo-cortical loops that control voluntary movement [43,44]. The direct output pathway (referred to as the ‘Go’ pathway) comprises spiny projection neurons (SPNs) that express dopamine D1 receptors and substance P [12,43,45]. The indirect output pathway (referred to as the ‘NoGo’ pathway) expresses dopamine D2 receptors and preproenkephalin [12,43,45]. Importantly, A_2A_ receptors are located predominantly on the axon terminal and cell of the ‘NoGo’ pathway which projects from the striatum to the GPe, and they are segregated from the ‘Go’ pathway [12,13,46,47,48].

This unique targeted distribution is key to understanding the potential of A_2A_ receptor antagonism in the management of PD because they are uniquely positioned to selectively modulate the activity of the ‘NoGo’ motor pathway at the level of the striatum and GPe [49,50]. Gs-coupled A_2A_ receptors are positively coupled to AC [7] and thus to the 3′,5′-cyclic adenosine monophosphate (cAMP)/protein kinase A (PKA) pathway [7], whose activation facilitates GABA release in the GPe [14,51]. An A_2A_ receptor agonist (CGS21680) or an AC activator significantly increased cAMP accumulation, subsequently activating the frequency of miniature inhibitory postsynaptic currents (mIPSCs) in GP slices [49]. CGS21680-induced cAMP accumulation was blocked by the adenosine A_2A_ receptor selective antagonist KF17837, and CGS21680-induced enhancement of the mIPSCs frequency, which are striatal GABAergic outputs, was inhibited by AC or PKA inhibitors. Also, in the striatum, adenosine A_2A_ receptor-induced inhibition of intrastriatal recurrent GABAergic activity, which cause excitation of the SPN cell body, has been revealed, and this phenomenon has been mimicked by cAMP analogues [52]. As a consequence, the entire GABAergic output activity (i.e., GABA release) of the SPNs is regulated by adenosine A_2A_ receptor mediated signaling through cAMP/PKA-dependent mechanisms [12].

In PD, the progressive loss of nigrostriatal neurons causes a decline in dopaminergic transmission with the indirect pathway to GPe becoming overactive [15,16], resulting in an excessive ’NoGo’ signal from the GPe to the thalamus. A_2A_ receptor activation also enhances the ‘NoGo’ signal while D_2_ receptor activation reduces the ‘NoGo’ signal [12]. While there remains much debate about the role of A_2A_-D_2_ chimeric receptors in modulating D_2_ receptor function [12,20,53,54], adenosine activation of A_2A_ receptors opposes the effects of drugs that act through D_2_ receptors (levodopa, dopamine agonists) that are used to relieve the motor symptoms of PD, acting as a ‘brake’ on their effects. This implies that an adenosine A_2A_ receptor antagonist should act to ‘release the brake’ and to increase the effects of dopaminergic medications (Figure 1). This is demonstrated in experimental models of PD by the ability of A_2A_ antagonism to increase the improvement in motor performance produced by levodopa and dopamine agonists in 6-OHDA lesioned rats and in MPTP-treated primates [55,56,57].

The direct pathway expresses dopamine D1 receptors and relays motor signals from the dopaminergic neurons of the striatum through the globus pallidus internal (GPi) and substantia nigra pars reticulata (SNr) to relieve the inhibition of the thalamus and signal to the motor cortex to initiate movement. In this way, dopamine signaling in the direct pathway functions like the “gas” powering a car. The indirect pathway functions to control unwanted movement and functions like the “brake” in a car. Neurons of the indirect pathway express both dopamine D2 receptors as well as adenosine A2A receptors. Upon activation of the indirect pathway, signals relayed through the globus pallidus external (GPe) to the GPi/SNR complex via the subthalamic nucleus leads to thalamic inhibition and reduction in the motor output signals to the motor cortex. For the indirect pathway, dopamine D2 receptors suppress activation; in contrast, adenosine A2A receptors activate the pathway. In Parkinson’s disease, dopamine signaling to both the direct and indirect pathway is reduced, causing slow and dysregulated movements Furthermore, abnormal activation of the indirect pathway can be enhanced via adenosine A2A receptors overexpressed in progression of the disease, causing further dysfunction of movement.

In addition, the abundant expression of A_2A_ receptors on the striatopallidal SPN [17] means that they are also uniquely positioned to integrate cortical glutamate signals and nigrostriatal/cortical dopamine signals to control striatal synaptic plasticity through a post-synaptic functional interaction with dopamine D_2_ receptors [58,59,60] and NMDA receptors [61,62], metabotropic glutamate 5 receptors [63,64,65], cannabinoid CB1 receptors [66,67], and presynaptic modulation of glutamate release at cortico-striatal projection terminals [68,69,70]. This complexity of activity may also contribute to the ability of adenosine A_2A_ receptors to modulate motor function but also to exert actions relevant to non-motor symptoms of PD, such as cognition, which will be explored in subsequent sections of this review [71,72].

One other component of the expression of A_2A_ receptors relevant to PD that requires mentioning is the increase in A_2A_ receptor expression seen in the caudate nucleus/putamen and GPe of those with the disease at post-mortem and also in patients by PET imaging [71,73,74]. This upregulation of A_2A_ receptor expression appears to occur early in the course of the illness, as it was found in the putamen of patients with Braak stage 1–2 disease [75]. The increased receptor expression was also more marked in those patients exhibiting dyskinesia. While it remains to be elucidated whether increased A_2A_ receptor expression is a cause or effect of dyskinesia and/or drug treatment, these data imply that some functional change has occurred in adenosinergic signaling in the basal ganglia, and this is reflected in some experimental studies [12,76,77]. Dopamine depletion by mitochondrial neurotoxins such as 6-OHDA and MPTP increased ATP release, upregulated ecto-5′-nucleotidase (CD73) and A_2A_ receptors in striatal synaptosomes [78] and in the striatum of MPTP-treated mice [79]. While the precise interpretation of these findings is not clear, preclinical data argue for an intervention in PD sooner rather than later using a selective adenosine A_2A_ antagonist [80,81].

### 2.2. Relevance of A_2A_ Receptor State to Mechanism of Action of Adenosinergic Agents in Managing the Motor Symptoms of Parkinson’s Disease

As discussed above, the therapeutic concept of A_2A_ receptor antagonists developed to date is through receptor blockade to release D_2_ function suppressing ‘NoGo’ signals of the SPN. However, it is now increasingly understood that interpretation of the mode of action for A_2A_ receptor agents in PD therapy should be discussed in terms of the receptor structure and formation to differentiate agonists, antagonists, and inverse agonists [81,82]. Several crystal structures of adenosine A_2A_ receptors have been resolved in a high-affinity antagonist (ZM241385)-bound state [83] and in the agonist (UK-432097)-bound state [84]. Studies show that A_2A_ agonists bind to the ligand binding pocket at the extracellular surface of adenosine A_2A_ receptors where there are hydrophobic side chains required for ligand recognition) inside the third and seventh transmembrane helical domains (H3 and H7) [84]. Three distinct activation states of adenosine A_2A_ receptors have been reported: the inactive state [83], the intermediately active state [84], and the active state [23]. Adenosine binds to the intermediate state and activates the adenosine A_2A_ receptor through interactions with H3 and H7 (Ser277, His278) in the ligand-binding pocket [82]. Inverse agonists (e.g., ZM241385) reduce basal activity by binding to the inactive form of the adenosine A_2A_ receptor [81]. Inverse agonists do not interact with H3 and H7; rather, they sterically prevent the conformational change in H5, thereby locking the receptor in its inactive state [81,82,85]. By contrast, neutral antagonists bind to the active state conformation of the adenosine A_2A_ receptor and suppress the agonist binding without any structural hindrances, [83] and allosteric modulators bind to sites outside of the orthosteric ligand-binding site [81,82,85,86]. Recently, the crystal structure of adenosine A_2A_ receptors bound to a partial agonist has also been resolved [87].

Adenosine A_2A_ receptors are considered to exist in equilibrium between active and inactive forms in the two-state model of GPCRs [81]. (Figure 2). Whereas endogenous adenosine and A_2A_ receptor agonists bind with higher affinity to the active form, inverse agonists have higher affinity for the inactive form—pushing the equilibrium towards the inactive state [12,76,77,80,81]. In the inactive state, adenosine A_2A_ receptors possess constitutive activity even in the absence of adenosine [81] and A_2A_ receptor inverse agonists act to further reduce cAMP accumulation generated by constitutive adenosine A_2A_ receptor activity [12,81]. By contrast, neutral antagonists bind to active forms of the adenosine A_2A_ receptor, thereby competitively antagonizing the effects of adenosine [12,81]. Thus, A_2A_ receptor antagonists such as istradefylline act in PD by modulating the D2 receptor-bearing indirect pathway activity [12,81]. By contrast, inverse agonism not only releases D_2_ function but also directly affects neuronal activity in the same direction as the D_2_ receptor, manifesting as further reduced cAMP production in the indirect pathway [88]. Again, it is the functional interpretation of these data that is important. They suggest that, while an adenosine, A_2A_ antagonist will antagonize the braking effect of adenosine on motor control via suppression of cAMP-dependent ‘NoGo’ pathway activity, it will not remove the constitutive receptor activity, which would continue to oppose the action of drugs acting through D_2_ dopamine receptors. In contrast, an inverse agonist would remove both the effects of adenosine on the A_2A_ receptor and its constitutive activity, having greater efficacy in normalizing the activity of the ‘NoGo’ pathway in PD by further suppressing basal levels of pallidal GABA release and also by enhancing dopaminergic drug action. This all suggests that an inverse agonist would be more likely to have an effect on motor dysfunction in PD when used as monotherapy than occurs with a classical antagonist, such as istradefylline [12]. This idea is supported by a recent proof-of-concept study of the antagonist KW-6356, which also shows inverse agonist activity [89,90].

Endogenous ligand adenosine binds with higher affinity to the active form of Gs-coupled A_2A_ receptors, and activates cAMP/PKA pathway via AC. This adenosine-induced cAMP accumulation activates PKA, following activation of multiple downstream targets, such as aPKC, CREB, and uORF5. Neutral antagonists bind to active forms of the adenosine A_2A_ receptor, thereby competitively antagonizing adenosine binding. In the inactive state, adenosine A_2A_ receptors possess constitutive activity with basal level production in cAMP, even in the absence of adenosine binding. Inverse agonists have higher affinity for the inactive form of the adenosine A_2A_ receptor than the active—pushing the equilibrium towards the inactive state. Thus, inverse agonists act to further reduce cAMP accumulation from the basal level generated by constitutive adenosine A_2A_ receptor activity. In addition, inverse agonists can also antagonize the effects of adenosine. A_2A_R: adenosine A_2A_ receptor, AC: adenylate cyclase, aPKC: atypical protein kinase C, cAMP: 3′,5′-cyclic adenosine monophosphate, CREB: cAMP response element-binding protein, GABA: γ-aminobutyric acid, PKA: protein kinase A, uORF5: upstream open reading frame 5 + indicates activation of signal transduction in absence of ligand, indicates suppression of constitutive activity by inverse agonist.

## 3. Clinical Actions of Adenosine A_2A_ Antagonists in Parkinson’s Disease

### 3.1. Motor Symptoms

The underlying science base for an effect of adenosine A_2A_ antagonists on the motor symptoms of PD is strong and is based on the selective localization of the receptor and its functional significance to basal ganglia activity [12,20]. Taken along with the evidence from animal models of PD, an improvement of motor disability would be predicted when adenosine A_2A_ antagonists are used in combination with levodopa therapy [32,91]. Not surprisingly, a number of selective adenosine A_2A_ receptor antagonists have been developed [92] with the aim of increasing the duration of motor improvement seen during the time when symptoms are adequately controlled by medication (ON time) and decreasing the periods of inadequate control of PD symptoms (OFF time) in patients receiving levodopa but where insufficient efficacy is occurring [18,32,93]. Surprisingly, however, clinical development has proved challenging. Two candidate molecules, vipadenant and BIIB 014, have undergone only limited human testing [94,95,96], with vipadenant development discontinued due to toxicological concerns. Three other compounds—istradefylline [97,98,99,100,101,102,103,104], preladenant [105,106] and tozadenant [107]—have been examined in phase II and phase III clinical trials, and while positive efficacy results were obtained in all phase II studies, tozadenant was discontinued due to toxicity concerns [108], and only istradefylline has successfully reached commercialization [18]. To date, istradefylline has received approval in the US and Japan for the treatment of adult PD patients experiencing OFF time who are currently taking levodopa (plus a decarboxylase inhibitor) [109,110]. The efficacy of istradefylline was evaluated in a pooled analysis of eight multicenter, phase IIb/III double blind, placebo-controlled trials of patients experiencing motor fluctuations taking 20 or 40 mg/day of istradefylline [93]. The overall effect size in reducing OFF time (−0.38 to −0.82 h) was in the same range as those achieved by other adjuncts to levodopa therapy—such as MAO-B inhibitors and COMT-inhibitors [93].

There are many reasons why the clinical studies of istradefylline and other adenosine antagonists has been challenging—for example, trial design, trial population selection, placebo effects—and these have been discussed these in detail elsewhere [18,19]. The concepts of ON and OFF time and other motor endpoints are tailored to reflect the effects of dopaminergic replacement therapy, and there are no clinical endpoints available that fully capture the effects of non-dopaminergic approaches. It has also been suggested that the population of PD patients studied is an issue that may be key to the problems encountered in clinical development [19]. Istradefylline was taken into clinical trials mainly using later stage patients with PD experiencing significant ‘wearing off’ time but who were already treated with optimized dopaminergic therapy—in other words, further improvement could not be achieved using levodopa, dopamine agonists, MAO-B inhibitors or COMT inhibitors. Thus, the effects observed were ‘in addition’ to dopaminergic medications, reflecting the non-dopaminergic nature of the drug in an already difficult to treat patient population. It is important to stress that all the preclinical science shows that istradefylline has its maximal effect in models of PD when used in combination with low or threshold doses of levodopa and shows little increase in motor performance when used with optimized, high doses of levodopa. This suggests that a new generation of adenosine A_2a_ receptor antagonists should be used at an earlier point in the progression of PD and in a manner where they are employed to be ‘levodopa sparing’ by avoiding further increases in levodopa dosage when the drug starts to exhibit a loss of efficacy. The other question that needs to be addressed is whether adenosine A_2A_ antagonists would be effective as early monotherapy, and this has received little attention so far [111,112]. As discussed above, it may be more relevant to utilize an antagonist/inverse agonist for these studies as and when one becomes available for use in PD [113,114,115].

### 3.2. Non-Motor Symptoms

Non-motor symptoms of PD are a major area of unmet need in treating the illness and probably outweigh the importance of further improvements in the control of motor function [116,117]. A_2A_ adenosine receptors are present in key ‘non-motor’ areas of the brain (e.g., nucleus accumbens, olfactory tubercle, amygdala, hippocampus, hypothalamus, thalamus, and cerebellum [17]) and play a role in the pathophysiology of sleep disturbances and mood disorders such as depression and anxiety which are common non-motor features of pre-motor and both early and late-stage clinical PD [28,118]. Importantly, preclinical investigation has delineated a role for adenosine signaling in attempting to address the development of novel approaches to their treatment [119,120,121]. To illustrate the progress being made, we will look at studies relevant to sleep–wake cycles, mood and cognition [119,120,121].

#### 3.2.1. Role of A_2A_ Receptors in Sleep–Wake Cycle Disturbance

Alterations to the sleep–wake cycle are a common non-motor symptom in PD and one that is often exacerbated by the use of dopaminergic therapies [122]. Nocturnal sleep disturbance is the most frequent manifestation, but excessive daytime somnolence also commonly occurs. Adenosine is a potent endogenous regulator of the sleep–wake cycle with adenosine levels increasing in the brain during periods of prolonged wakefulness and decreasing during sleep [123]. Excessive daytime somnolence in PD may be a result of over-activation of A_2A_ receptors in the nucleus accumbens [32]. The potential for the use of adenosine antagonists in controlling disturbances in the sleep–wake cycle is shown by looking at the arousal effects of caffeine, a non-selective adenosine receptor antagonist. These are clearly mediated through the A_2A_ receptor as caffeine’s ability to induce wakefulness is lost when A_2A_ receptors are inactivated or deleted in the nucleus accumbens of mice but not when A_1_ receptor function is disrupted [120,122,124].

Two small open-label studies of istradefylline have investigated its impact on excessive daytime somnolence and nocturnal sleep. The first, an open-label study, evaluated patients over 12 weeks using the Epworth Sleepiness Scale (ESS), PD sleep scale (PDSS)-2, and PD Questionnaire (PDQ-8) at baseline and over a three-month period [125]. The study found that, in addition to improving motor scores, istradefylline decreased ESS scores; however, no impact on the PDSS-2 and PDQ-8 score were observed [125]. In a second study assessing istradefylline use over a month, patients taking istradefylline exhibited a decrease in ESS without a change in the total PD Sleep Scale (PDSS) [126]. These findings show that istradefylline may have an impact on sleep disturbances in PD patients, though additional large-scale studies are needed.

#### 3.2.2. Role of A_2A_ Receptors in Mood Disorders

Mood disorders are a common feature of PD, with depression affecting at least 35% of the patient population [111,112,127]. Depression can be secondary to the changes in quality of life brought on by a diagnosis of PD but may also occur directly due to the progression of pathological change that occurs as the illness advances [128,129]. Loss of dopaminergic input from the substantia nigra to the caudate nucleus and putamen disrupts the activity of the striato-thalamo-cortical loops, altering not only motor function but also the affective pathways controlling limbic and cortical function [32,130].

A role for A_2A_ receptor activity in mediating symptoms of depression has been demonstrated using rodent models of depression (the forced swim test, tail suspension tests and learned helplessness) and examining the effects of adenosine A_2A_ antagonist drugs [32,131,132]. For example, istradefylline decreased the immobility time in both the forced swim test and tail suspension tests [131]. When istradefylline was co-administered with fluoxetine, an additive effect was observed in reversing depressive behaviors [131]. Acute as well as chronic oral administration of istradefylline improved depression-like behavior in the rat learned-helplessness model [132]. The role of A_2A_ receptors in anxiety in rodent models has also been investigated but the results are difficult to interpret [32]. A_2A_ receptor knockout mice exhibit anxiety-like behaviors, but the effects seem dependent on the brain area investigated [133]. Some studies also report that A_2A_ antagonists are able to ameliorate anxiety-like behaviors in rodents [117]. This is clearly an area relevant to mood changes in PD and where an adenosine A_2A_ antagonist may be useful, but it is one that requires further study both at the preclinical level and evaluation in man.

Two small open-label studies have evaluated the efficacy of istradefylline for the treatment of depression or mood disorder in PD and are suggestive of some improvement. Patients with motor fluctuations not taking an antidepressant at baseline were evaluated using the Patient Health Questionnaire (PHQ) for depression. Eleven of fourteen patients had at least mild depression at baseline, and after 8 weeks of istradefylline (20–40 mg/day) treatment, five of fourteen patients showed an improvement in PHQ scores, with a trend to better improvement in those with worse (higher) PHQ scores at baseline [134]. In a second study, 30 patients with depressive symptoms not taking an antidepressant were studied at baseline and again after 12 weeks on istradefylline (20–40 mg/day). Significant improvement was seen in the Snaith–Hamilton Pleasure Scale Japanese version (SHAPS-J), Apathy Scale, and Beck Depression Inventory—2nd edition (BDI) [135]. Interestingly, while motor symptoms also improved with istradefylline administration, this was not correlated with the relief of depressive symptoms, indicating that A_2A_ antagonism had an ameliorative effect on mood disorders independent of its effect on motor activity.

### 3.3. Role of A_2A_ Receptors in Cognitive Impairment

There are multiple longitudinal studies showing an inverse relationship between consumption of caffeine and decreased memory impairments associated with aging as well as a reduced risk of developing dementia and AD [136,137,138,139,140,141]. Plasma levels of caffeine in subjects with mild cognitive impairment (MCI) who later progressed to dementia were lower than those whose cognition remained stable [142,143]. Furthermore, in a cross-sectional study in drug naïve PD patients, coffee drinking was significantly associated with a reduced severity of the mood/cognition domain of the non-motor symptom scale (NMSS) [144]. Accumulating evidence from preclinical studies in normal rodents and non-human primates also demonstrate that A_2A_ receptor antagonists can enhance working memory [145], reversal learning [146], set-shifting [147], goal-directed behavior [148], and Pavlovian conditioning [133]. Consistent with increased A_2A_ receptor expression that occurs in the brain in PD, recent studies have demonstrated that optogenetic stimulation of A_2A_ receptor signaling [116] or transgenic overexpression of hippocampal A_2A_ receptors [149] was sufficient to drive memory impairments in young animals. Overexpression also triggers HPA-axis dysfunction and a reduction in hippocampal glucocorticoid receptor levels [150].

Cognitive change occurs in both early PD impacting executive function (e.g., planning and decision making), working memory and procedural learning [151,152], and in late-stage illness with cognitive decline and frank dementia [152,153]. Early cognitive impairment in PD may relate to the dysfunction of the dopaminergic system and loss of LTP [29,154]. Dopamine replacement therapy may improve some aspects of cognition but not all while over-dosage may worsen cognition [155,156,157]. Dementia in PD is currently treated with cholinesterase inhibitors such as rivastigmine that provide only modest benefit [29,154]. Clearly there is a need for other pharmacological approaches to treatment, and epidemiological studies of caffeine consumption in aging populations suggest a role for adenosine receptors.

In rats, dopamine depletion in the prefrontal cortex (PFC) produces cognitive deficits, particularly associated with executive dysfunction [158,159], visuo-spatial, and non-spatial working memory that resemble those seen in PD [160]. Istradefylline reversed the cognitive deficits, and this was associated with increased dopamine levels in the PFC [161]. Istradefylline also improved spatial working memory in the MPTP-treated primate model, providing further preclinical data to suggest a potential benefit in man [162]. In rodents, A_2A_ receptor antagonists reversed the deficits in working memory and motor sequence learning induced by the intracerebral injection of α-synuclein [163,164]—a protein species closely associated with neuronal toxicity in PD. Similarly, impairment of LTP associated with α-synuclein exposure was reversed in A_2A_ receptor knockout mice or following A_2A_ receptor blockade through an NMDA receptor-dependent mechanism.

Collectively, there is a convergence of clinical, epidemiological and experimental evidence supporting the A_2A_ receptor as a potential therapeutic target for improving cognitive impairments in both AD and PD.

## 4. Potential Neuroprotective Effects of A_2A_ Receptor Antagonism in PD

Uncovering a neuroprotective or disease modifying therapy for PD equates to the search for the holy grail. The promising activity of a range of compounds in a wide spectrum of preclinical models of the illness has stubbornly refused to translate into a clinical effect that meets with regulatory approval. Adenosine A_2A_ receptors may provide another tantalizing lead to follow based on epidemiological data and findings from laboratory studies investigating the protective effects A_2A_ receptor knockout and A_2A_ antagonists against a variety of toxic insults to dopaminergic neurons.

The initial clue for the potential neuroprotective effect of A_2A_ receptor antagonists came from a prospective epidemiological study (The Honolulu Heart Program) carried out over a 30-year period. Daily consumption of 784 mg/day or more of coffee during mid-life was shown to reduce the risk for developing PD at the age of 65 by fivefold compared to non-coffee drinkers after age- and smoking adjustments [165]. Since then, systemic analysis from over 20 observational studies has firmly established that regular human consumption of caffeine is associated with a fivefold reduction in risk for developing PD [165,166,167,168,169]. A meta-analysis of 13 studies, involving over 900,000 participants, found a non-linear relationship between coffee intake and the risk of PD with maximum protection seen at approximately three cups per day [170]. More recently, the Harvard Biomarkers Study conducted a cross-sectional, case–control study which confirmed the robustness of lower caffeine intake as a factor inversely associated with the risk of PD [171].

While these data sound compelling, there are several important caveats on this inverse relationship between coffee consumption and risk of developing PD which need to be considered:
(1)The effect may not be due to caffeine—however, the consumption of decaffeinated coffee was not associated with a reduced risk of developing PD in the Health Professional Follow-up Study [166].(2)The effect is not universal in the population—it is strong and consistent in men in the Health Professional Follow-up Study [169] and in post-menopausal women who have never used hormone replacement therapy in the Cancer Prevention Study II Nutrition Cohort but uncertain in women and post-menopausal women who have used hormone replacement therapy at some point [172].(3)The effect could be due to early premotor features of PD causing a reduction in caffeine consumption—however, the persistence of these predictive associations in lag analyses that exclude PD cases that occurred within 2–6 years from dietary survey suggests this is unlikely.(4)The effect may reflect genetic predisposition to PD—there is emerging evidence that links dietary [173,174,175] and plasma [176]) caffeine exposure as being more negatively associated with the likelihood of developing PD among carriers of a pathogenic *LRRK2* mutation than among PD patients who do not carry *LRRK2* mutation [173,176].

The neuroprotective potential of A_2A_ receptor antagonists is further substantiated by mounting evidence from experimental studies that demonstrate A_2A_ receptor blockade confers protection against the degeneration of dopaminergic neurons induced by the toxins, MPTP, 6-OHDA and rotenone [24,177,178,179,180,181,182] and by intracerebral injection of toxic α-synuclein particles [183]. In studies using caffeine, protection against MPTP-induced dopaminergic toxicity in mice was observed after either acute or chronic treatment [24,178,184,185], and importantly, when caffeine was administered after the onset of the neurodegenerative process, reflecting how it might be used in a clinical setting [186]. A_2A_ receptor knock out also prevented the loss of dopaminergic neurons caused by the intra-striatal injection of preformed A53T α-synuclein fibrils or by transgenic overexpression of the human A53T and A30P α-synuclein mutations associated with familial forms of PD [187].

These animal investigations provide a neurobiological basis for the inverse relationship between caffeine consumption and the reduced risk of developing PD established by the epidemiological studies. Collectively, these findings define aberrant A_2A_ receptor signaling as a critical pathogenic mechanism of α-synuclein-triggered neurodegeneration in PD models and support the clinical potential for caffeine and A_2A_ receptor antagonists as potential disease-modifying approaches to treatment. US approval and availability of istradefylline opens up a new opportunity for long-term phase IV clinical studies and/or real-world data analysis of istradefylline in PD that monitor the course of the illness and evaluate the potential disease modifying effects of istradefylline.

## 5. Summary and Conclusions

It is now understood that PD is more than a motor disorder, and the effects of A_2A_ antagonism on the non-motor symptoms of this multifactorial disease are likely to take greater prominence in treatment decisions. Just as importantly, adenosine antagonism avoids the inevitability of the dopaminergic side effect profile that currently limits the classic dopamine replacement therapies. The potential for neuroprotection also adds another important element that requires further exploration. There is now momentum to understand the role of A_2A_ receptors in a variety of other neurodegenerative diseases (Box 1) and its selective localization provides an attractive therapeutic target. Increased A_2A_ receptor expression has been reported in the brain of patients with AD, amyotrophic lateral sclerosis, Huntington’s disease and multiple sclerosis [188,189,190,191,192,193,194,195].

In addition, A_2A_ receptor antagonists show beneficial effects in animal models of each neurodegenerative disease. Within this context, the A_2A_ receptor has become a seminal example of utilizing high-resolution structures of GPCRs for the rational design of diverse drug molecules. The introduction of istradefylline as the first adenosinergic drug for use in PD opened a new era in the pharmacological treatment of the illness. The continued development of new A_2A_ receptor compounds (partial and inverse agonists as well as antagonists) opens exciting avenues for neurodegenerative disease management.

Box 1Adenosine A_2A_ receptors in other neurological diseases.Interest in A_2A_ receptors as a therapeutic target for other neurodegenerative diseases including Alzheimer’s disease (AD), amyotrophic lateral sclerosis (ALS), Huntington’s disease (HD) and multiple sclerosis (MS) has grown exponentially in recent years. Both agonists and antagonists to this receptor have been investigated in the respective animal models of disease and support a potential role for pharmacological manipulation of A_2A_ receptors in each of the pathologies.
**Alzheimer’s Disease (AD)**
A_2A_ receptors are increased in both patients and animal models of AD [188,189,190,191].Istradefylline reduces memory deficits in aging mice with amyloid pathology [196].Age-related shift in LTD is dependent on neuronal A_2A_ receptors [189].A_2A_ receptor knockout mice slow the decrease in Aβ/Tau toxicity [188,197].

**Amyotrophic Lateral Sclerosis (ALS)**
A_2A_ receptors are increased in the spinal cord of patients [192].A_2A_ receptors are increased at the early symptomatic stage in a mouse model of ALS [192], but decreased at the end stage [198].Istradefylline significantly delayed disease progression [192] and rescued LTP impairment adenosine A_2A_ receptors [199] in the mouse model of ALS.Partial genetic ablation of A_2A_ receptors significantly delayed disease progression in SOD1G93A mice [192].The timing by which the alterations of the adenosinergic system occur during ALS pathogenesis in patients and animal models is a key factor to completely understand its contribution to disease progression and to identify the proper therapeutic window for putative treatments.

**Huntington’s disease (HD)**
A_2A_ receptors are decreased in the striatum of HD patients [200].There is a 50% reduction in A_2A_ receptor striatal binding in patients with HD [201].A_2A_ receptors are decreased at the presymptomatic and symptomatic stage in animal models of HD [200,202].A reduction in radiotracer binding for A_2A_ receptors is found in the lesioned side compared with the intact side in a rat model of HD [203].Several A_2A_ receptor antagonists show multiple beneficial effects in chemical- and lesion-induced HD models [204,205].A_2A_ receptor knockout mice have worse survival and motor behaviors in the Tg mouse model of HD [202].However, the effect of A_2A_ receptors blockade in HD mice appears to be complex and could be detrimental because it compromises the function of BDNF [202].

**Multiple sclerosis (MS)**
A_2A_ receptors are increased in lymphocytes from patients with MS and animal models [193,194].A_2A_ receptors are increased in the brain of secondary progressive MS [195].A_2A_ receptors are highly expressed on infiltrating macrophages [206].A_2A_ receptor antagonist (SCH58261) protects against experimental autoimmune encephalomyelitis (EAE) in mice [207].A_2A_ receptor agonist (CGS21680) inhibits the EAE progression in mice [208].A_2A_ receptor knockout mice show more severe EAE pathology and neurological deficits in EAE model mice [209].

**Summary of adenosine A2A receptors and neurological disease**


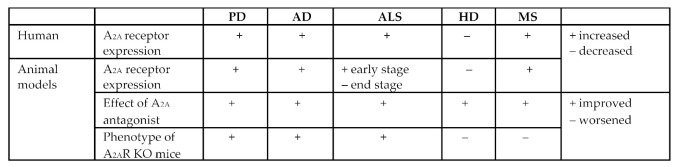



We dedicate this paper to the contribution of Dr Kenneth Jacobson to the field of adenosine receptor science. Over a decades-long and distinctive research career at NIH, Dr. Kenneth Jacobson has led the pioneering work in design, chemically synthesizing, and characterizing pharmacologically new agonists and antagonists for all four subtypes of adenosine receptors, including A_2A_ receptor agonists and antagonists. Through homology to known structures (X-ray and cryo-electron microscopy), Dr. Jacobson contributed to exploration and determination of three-dimensional structures of the A_2A_ receptor (in both agonist- and antagonist-bound states). Dr. Jacobson’s lab has also designed, synthesized and provided functionalized congeners of inhibitors (or antagonists) of the A_2A_ receptor and the enzyme CD73, of adenosine receptor ligands coupling to nanoparticles, such as quantum dots and gold particles, for tissue-specific targeting.

## Figures and Tables

**Figure 1 molecules-27-02366-f001:**
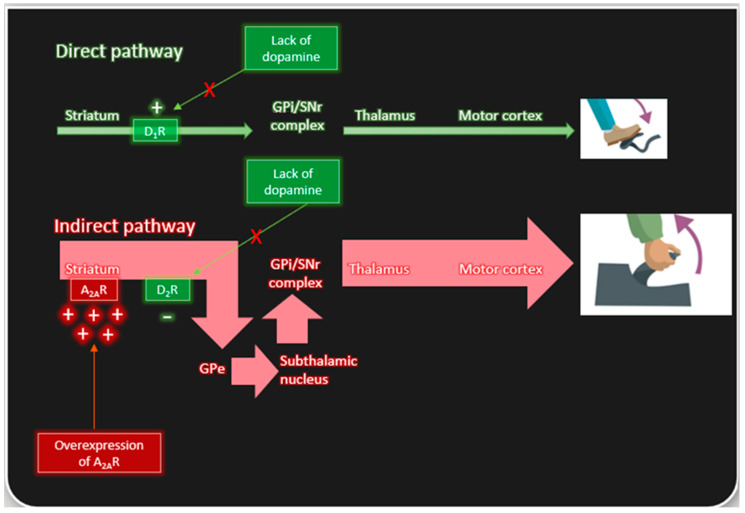
The direct and indirect pathways work together to control movement.

**Figure 2 molecules-27-02366-f002:**
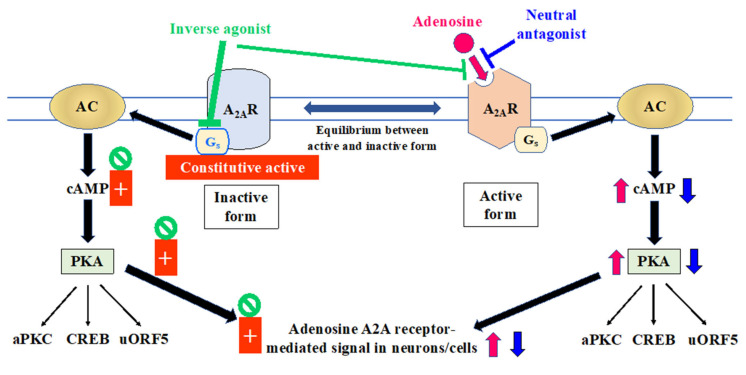
Two-state model of adenosine A_2A_ receptor and its major intracellular signal transduction.

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
