# Peer review of "The Pharmacological Potential of Adenosine A_2A_ Receptor Antagonists for Treating Parkinson’s Disease"

_molecules, 2022, doi:10.3390/molecules27072366_

Round 1
Reviewer 1 Report
In this manuscript, the authors reviewed the pharmacology of adenosine A2A receptor antagonists from the perspective of the treatment of both motor and non-motor symptoms of PD and their potential for disease modification. In addition, they described the role of A2A receptors as a therapeutic target for other neurodegenerative diseases including Alzheimer’s disease, amyotrophic lateral sclerosis, Huntington’s disease and multiple sclerosis.
This review is interesting and well-written; unfortunately, the manuscript needs substantial improvements and corrections before publishing may be possible. As a review, it should give a more or less complete overview of the literature.
General points:
Please add a list of abbreviations before the References section to your manuscript.
For better readability, please add more Figures to your manuscript.
Special points:
Keywords
Please add also to keywords: non-dopaminergic pharmacological target.
Introduction
The Introduction section should be improved, i. e., by substantial references in the field:
Lines 22-27: please add multiple references at the end of each of these sentences.
Lines 29-31: please add multiple references at the end of this sentence.
Lines 32-34: please add more references at the end of this sentence.
Lines 36-44: please add multiple references at the end of each of these sentences.
Lines 44-46: please add multiple references at the end of this sentence.
- Adenosine A2A receptors and Parkinson’s disease
Lines 53-71: please add multiple references at the end of each of these sentences.
Lines 73-76: please add more references at the end of this sentence.
2.2. Basal ganglia expression of A2A receptors
Lines 81-86: please add multiple references at the end of each of these sentences.
Lines 86-88: please add more references at the end of this sentence.
Lines 88-89: please add multiple references at the end of this sentence.
Lines 101-103: please add multiple references at the end of this sentence.
Lines 137-141: please add multiple references at the end of each of these sentences.
2.3. Relevance of A2A receptor state to mechanism of action of adenosinergic agents in managing 146 the motor symptoms of Parkinson’s disease
Lines 148-152: please add multiple references at the end of each of these sentences.
Lines 163-164: please add multiple references at the end of this sentence.
Lines 174-178: please add multiple references at the end of each of these sentences.
Lines 183-190: please add multiple references at the end of each these sentences.
Clinical actions of adenosine A2A antagonists in Parkinson’s disease
3.1. Motor symptoms
Lines 205-209: please add multiple references at the end of each of these sentences.
Lines 209-215: please add multiple references at the end of each of these sentences.
Lines 217-220: please add multiple references at the end of this sentence.
Lines 223-227: please add multiple references at the end of this sentence.
Lines 230-233: please add multiple references at the end of this sentence.
Lines 235-247: please add multiple references at the end of each of these sentences.
3.2. Non-Motor Symptoms
Lines 253-255: please add multiple references at the end of this sentence.
Lines 255-257: please add more references at the end of this sentence.
Lines 259-261: please add multiple references at the end of this sentence.
3.2.1. Role of A2A receptors in sleep-wake cycle disturbance
Lines 264-267: please add multiple references at the end of each of these sentences.
Lines 270-272: please add multiple references at the end of this sentence.
3.2.3. Role of A2A receptors in mood disorders
Lines 287-288: please add more references at the end of this sentence.
Lines 288-290: please add multiple references at the end of this sentence.
Lines 290-293: please add more references at the end of this sentence.
3.3. Role of A2A receptors in cognitive impairment
Lines 340-344: please add multiple references at the end of each of these sentences.
- Summary and conclusions
Lines 427-431: please add multiple references at the end of this sentence.
Lines 431-432: please add multiple references at the end of this sentence.
Box 1. Adenosine A2A receptors in other neurological diseases
Lines 451-456: please add multiple references at the end of each of these sentences.
Adenosine A2A receptors and neurological disease
Please add a title and Legend to this Table. In addition, please add to each line (near of “plus” or “minus”) the appropriate references and references numbers according to your List of references.
Legend Figure 1
Please add to this Legend a more detailed description.
Author Response
Please see attachment (for easier reading)

Reviewer 2 Report
The review “The Pharmacological Potential of Adenosine A2A Receptor Antagonists for Treating Parkinson’s Disease” by Mori et al., explores the role of Adenosine A2a receptor in PD. The review is well written and covers all aspects of PD associated with the A2a receptor.
The manuscript can be accepted for publication.
Author Response
We thank the reviewer for their kind comments.
Reviewer 3 Report
The paper by Mori et al., is an excellent review of the pharmacological evidence underlying the use of A2A adenosine receptor antagonists for the treatment of both motor and non-motor symptoms of Parkinson’s Disease, as well as their potential as disease-modifying therapy. The authors concisely summarize all the crucial information regarding the preclinical and clinical pharmacology of A2A adenosine receptor antagonists, supporting their role as a target not only for Parkinson’s Disease but also for other neurodegenerative diseases and neuropsychiatric disorders.
Author Response
We thank the reviewer for their kind comments
Reviewer 4 Report
I have carefully read the review by Mori et al. The paper offers a comprehensive overview of A2a receptor antagonists effects in Parkinosn's disease. The review is well organized and written and may be of interest for purinergic scientists.
Minor:
Please add the reference by Merighi et al., 2021 about expression of A2A receptors in brain and platelets of patients with Alzheimer's disease.
Author Response
We thank the reviewer for their kind comments, and have added the requested reference.
Round 2
Reviewer 1 Report
The authors did a really good job - all my critical points have been successfully adressed. I have no further suggestions or concerns.